# SECURE AUTOREGRESSIVE INFERENCE WITH PROMPT SEPARATION VIA KEY-VALUE CACHING

## ABSTRACT

Large Language Models (LLMs) have demonstrated remarkable performance, driving their widespread adoption across various applications. This prevalence increases the importance of user request privacy during inference. While Fully Homomorphic Encryption (FHE) and Secure Multi-Party Computation (MPC) offer promising solutions for privacy-preserving inference, they suffer from significant latency overhead, limiting practical deployment. Prior research has explored more efficient cryptographic primitives and polynomial approximations for non-linear operations. However, the inference latency remains significantly higher than that of plaintext execution. To further mitigate computational overhead, we introduce a novel approach that leverages prompt separation with key value caching. Our method accelerates secure inference by processing non-sensitive tokens in plaintext and using their key-value caches when subsequently processing private tokens. To ensure effective contextual reasoning, we also introduce an attention mask adjustment mechanism that constrains privacy-sensitive tokens to attend to nearby tokens from their original masked positions. Through experiments across various LLM architectures and MPC frameworks, we show that our approach achieves a 1.5-2.5× reduction in inference latency without significant performance degradation.

## 1 INTRODUCTION

Large Language Models (LLMs) have achieved remarkable progress in understanding and reasoning across diverse domains, including general knowledge, programming, and medicine (Achiam et al., 2023; Team et al., 2024; Singhal et al., 2023). As a result of these broad capabilities, individuals and organizations are increasingly integrating LLMs into their workflows to enhance productivity. However, LLMs with these strong reasoning capabilities often comprise hundreds of billions of parameters, making local execution impractical and necessitating cloud-based inference. Thus, clients must transmit requests to model providers, which introduces privacy leakage concerns. This issue has become particularly critical in the enterprise and medical domains, where prompts may contain sensitive information. For example, corporate queries may involve technical or managerial data, while medical applications often include sensitive personal medical data, such as diagnoses. These privacy issues represent a critical barrier to the broader adoption.

Secure computation approaches, including Fully Homomorphic Encryption (FHE) (Gentry, 2009) and Secure Multi-Party Computation (MPC) (Yao, 1986), are explored as promising solutions to this privacy issue. However, inherent limitations pose challenges when utilizing these approaches. First, non-linear operations are major efficiency bottlenecks, often requiring polynomial approximations that introduce approximation errors. Moreover, FHE requires considerable computation, while MPC incurs substantial communication costs. Since LLMs involve non-linear operations such as layer normalization (Ba, 2016), softmax, GELU, and SiLU (Hendrycks & Gimpel, 2016), and require intensive computation, these limitations significantly impede the effective deployment of FHE and MPC for secure LLM inference.

To overcome these challenges, prior works propose efficient approximations or cryptographic primitives. MPCFormer (Li et al., 2023) approximates non-linear operations using polynomials and utilizes knowledge distillation to mitigate the performance degradation introduced by these approximations. Iron (Hao et al., 2022) and Bolt (Pang et al., 2024) leverage a hybrid approach, utilizing FHE for linear operations and MPC for non-linear ones to exploit the strengths of each approach. More

recently, CipherPrune (Zhang et al., 2025b) introduces an orthogonal approach that progressively prunes input tokens during inference. Nevertheless, the inference latency of these secure approaches remains significantly greater than plaintext execution.

Motivated by the observation that not all tokens in a prompt are privacy-sensitive, we propose a novel approach to reduce the computational overhead of secure inference by prompt separation using key-value caching. Specifically, we identify sensitive tokens using a Personally Identifiable Information (PII) detector (Microsoft, 2024) (or as specified by clients) and mask them in the original prompt. We first compute the key-value cache for the masked prompt in plaintext, and then securely process the sensitive tokens using the cached values. To ensure that privacy-sensitive tokens attend to appropriate contextual information, we introduce an attention mask adjustment mechanism that encourages the model to focus on tokens near the original masked positions. Our approach is compatible with any autoregressive model and can be integrated with existing approximation techniques and cryptographic primitives. In our experiments, we demonstrate that our approach achieves a 1.5–2.5× reduction in inference latency without significant performance degradation on GPT2 (Radford et al., 2019) and Qwen2 (Yang et al., 2024) under two MPC frameworks, CrypTen (Knott et al., 2021) and SPU (Ma et al., 2023), across multiple subjects in the MMLU benchmark (Hendrycks et al., 2020).

Our contribution is threefold:

- We propose a novel and efficient secure inference method based on prompt separation, which precomputes key-value caches for non-sensitive tokens in plaintext to reduce secure computation.
- We introduce an attention mask adjustment mechanism that guides the model to attend to relevant contextual tokens when processing sensitive inputs.
- We empirically demonstrate that our method accelerates secure inference across multiple LLM architectures and MPC frameworks, achieving significant latency reduction without a significant performance drop.

## 2 PRELIMINARIES

### 2.1 AUTOREGRESSIVE LANGUAGE MODELS AND KEY-VALUE CACHING

Autoregressive language models progressively generate subsequent tokens conditioned on the input and previously generated tokens. This is because for a model $\theta$ and a sequence $\mathbf{x} = (x_1, \ldots, x_n)$, they model the joint probability of the token sequence as: $p_\theta(\mathbf{x}) = p_\theta(x_1) \prod_{i=2}^{n} p_\theta(x_j|x_1, \ldots, x_{i-1})$. Thus, for generated tokens $(x_1, \ldots, x_i)$, the model $\theta$ samples the next token from the distribution $p_\theta(x_{i+1}|x_1, \ldots, x_i)$. Although this autoregressive mechanism is effective, it demands significant computations that grow with the number of output tokens. To avoid redundant computations during each inference step, key-value caching is used. Specifically, within the attention mechanism, models utilize causal masks that restrict each token to only attend to preceding tokens. Consequently, when predicting the $(i+1)$-th token, the keys and values for $(x_1, \ldots, x_{i-1})$ are the same as when predicting the $(i)$-th token. By caching the keys and values of the preceding tokens, redundant computation can be avoided to predict the subsequent token. Specifically, when computing the attention of the $(i)$-th token, the query $Q_i$, key $K_i$, and value $V_i$ for the current token are calculated. The keys $[K_{1:i-1}]$ and values $[V_{1:i-1}]$ from the previous tokens are reused (cached). Then, the attention for the $(i)$-th token is computed as:

$$\text{Attention}(Q_i, [K_{1:i}], [V_{1:i}]) = \text{softmax}\left(\frac{Q_i[K_{1:i}]^\top + M_i}{\sqrt{d}}\right)[V_{1:i}], \tag{1}$$

where $[K_{1:i}]$ and $[V_{1:i}]$ denote the concatenation of the cached keys and values with the current token's key and value, respectively, and $d$ is the dimension of the head. Here, $M_i$ is a row vector representing the $(i)$-th row of the causal attention mask, which will have $0$ for the first $i$ elements and a large negative value for the elements after $i$. We note that, for decoder-only language models, including GPT (Radford et al., 2019), Llama (Touvron et al., 2023), and Qwen (Yang et al., 2024), the input prompts are also processed using causal masks. As a result, even within the input prompt, each token's key and value depend only on its predecessors, enabling their precomputation. Inspired by this inherent property, our approach precomputes non-sensitive tokens in the plaintext space and utilizes their key-value caches when computing privacy-sensitive tokens securely.

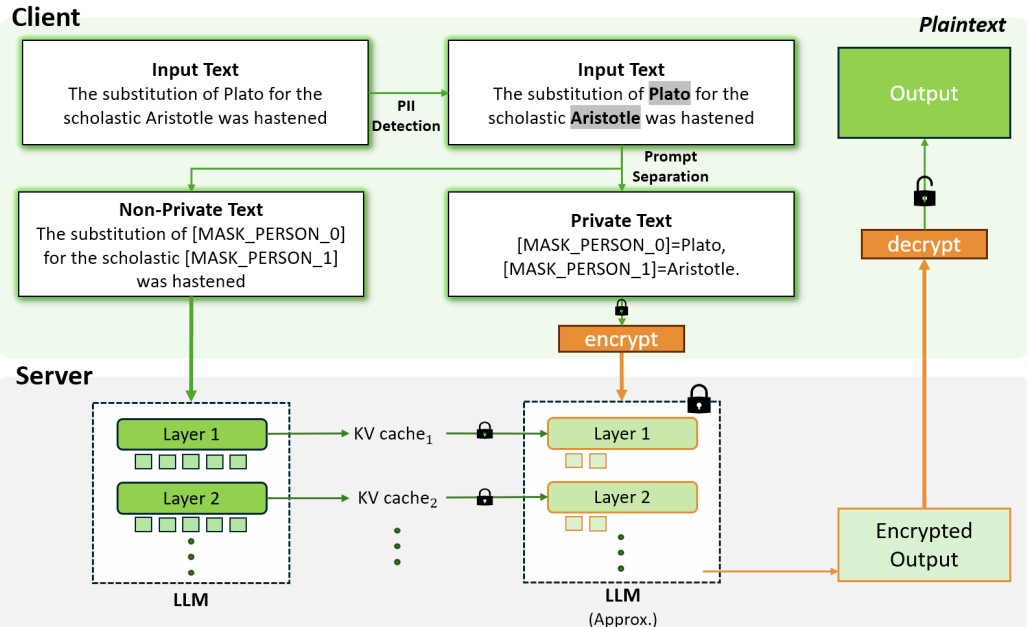

Figure 1: The overall pipeline of our method. The client first masks privacy-sensitive tokens in the prompt, separating it into a masked prompt and corresponding mask information. The masked prompt is sent to the server in plaintext, while the mask information is transmitted securely (via secret sharing or homomorphic encryption). The server computes key-value caches for the masked prompt in plaintext, then predicts subsequent tokens securely using the mask information and precomputed caches. Finally, the encrypted output is returned to the client, who decrypts it to obtain the result.

## 2.2 SECURE INFERENCE FOR LANGUAGE MODELS

Secure inference aims to enable correct inference without revealing the original values of inputs or outputs to the model provider, and without revealing model parameters to clients. This allows both clients and model providers to protect their sensitive information from each other. Fully Homomorphic Encryption (FHE) (Gentry, 2009; Cheon et al., 2017) and Secure Multi-Party Computation (MPC) (Yao, 1986; Evans et al., 2018) are commonly used for this purpose. FHE preserves privacy by encrypting the input prompt before sending it to the model provider, whereas MPC achieves privacy by splitting values into multiple shares, distributing them, and conducting inference collaboratively. Recovering the original values in both security systems is computationally infeasible. However, both approaches have a limitation: the latency of both systems is significantly higher than that of plaintext inference. FHE is slow due to bootstrapping and incurred operations from packing, while MPC suffers from substantial communication costs. Especially, the non-linear operation is a major efficiency bottleneck, so various methods have proposed more efficient approximations to maintain performance (Li et al., 2023; Dong et al., 2023; Zimerman et al., 2024; Zhang et al., 2025b). At the cryptographic level, hybrid protocols that leverage FHE for linear operations and MPC for non-linear operations have also been developed to improve efficiency (Reagen et al., 2021; Li et al., 2024; Lu et al., 2025). However, the inference speed remains significantly slower than in plaintext, requiring substantial acceleration for practical deployment. In this work, we propose an efficient inference mechanism that can be integrated with previous works to achieve further speedups by prompt separation with key-value caching.

## 3 PROPOSED METHOD

We now propose our method to reduce computational overhead in secure inference. Our intuition is that not all tokens in a prompt are privacy-sensitive, so we can decrease the computation required in secure inference by processing non-sensitive tokens of the input in plaintext. Specifically, the client first identifies privacy-sensitive tokens in the input prompt using a Personally Identifiable

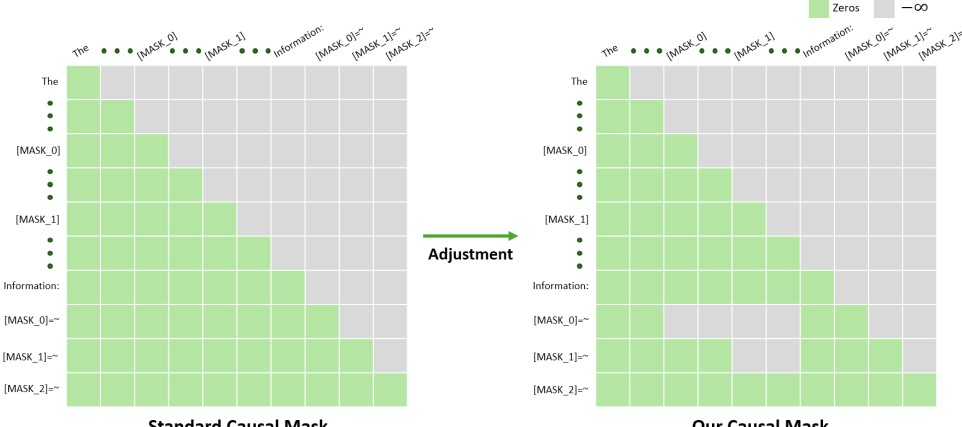

Figure 2: Comparison between the standard causal mask and our adjusted causal mask. Green blocks represent zero values, while gray blocks indicate large negative values. To encourage the model to attend to tokens near the original position of a masked phrase when processing the corresponding mask information, our adjusted mask assigns large negative values to tokens that appear after the original mask position in the masked prompt. For brevity, entity types are omitted.

Information (PII) detector (Microsoft, 2024) (or user specification), and these sensitive tokens are masked. The masked prompt is then sent to the model provider (the server) in plaintext, while the sensitive tokens are transmitted securely (via homomorphic encryption or secret sharing). The model provider computes the key-value (KV) cache for the masked prompt in plaintext and securely processes the sensitive tokens using the precomputed KV cache (as detailed in Section 3.1). To ensure that sensitive tokens consider appropriate contextual cues, we adjust the attention mask of privacy-sensitive tokens to prevent them from attending to positions beyond their original location, except for the mask information (in Section 3.2). The overall pipeline is provided in Figure 1.

## 3.1 PROMPT SEPARATION USING KEY-VALUE CACHING

To discriminate privacy-sensitive tokens, they are either specified directly by clients or detected automatically when unspecified. In the latter case, we use a Personal Identifiable Information (PII) identifier to detect privacy-sensitive phrases within the prompt. Note that PII detection has been extensively studied, and recent methods demonstrate near-perfect accuracy (Shen et al., 2025). For our experiments, we utilize the widely adopted open-source detector Presidio (Microsoft, 2024), but other detectors can also be used. Specifically, we treat phrases related with names, locations, dates, websites, URLs, phone numbers, and email addresses as privacy-sensitive. Note that our method only requires the specification of sensitive tokens, making it compatible with any PII detection framework.

After detecting privacy-sensitive phrases, we assign each identified phrase a numerical label based on its order of appearance from the beginning of the prompt. Each privacy-sensitive phrase is then replaced with a phrase of the form "[MASK_*type*_#]", where *type* indicates the entity category (such as name, location, and date), and # is the assigned label. Note that numbering is conducted independently for each entity type, so identical indices may appear across different categories. For example, given the prompt "The substitution of Plato for the scholastic Aristotle was hastened", the masked prompt becomes "The substitution of [MASK_PERSON_0] for the scholastic [MASK_PERSON_1] was hastened". This allows us to process the masked prompt in plaintext while preserving the privacy-sensitive content. To securely recover the original semantics, the mask information is appended to the end of the prompt in the format as "Masked information: [MASK_PERSON_0]=Plato, [MASK_PERSON_1]=Aristotle". These phrases are processed securely using the precomputed key-value cache of the masked prompt. To reduce the redundant masks, repeated occurrences of the same sensitive phrase are replaced with the same mask. Several examples of the original prompts and their corresponding masked versions with appended masked information on the benchmark dataset are provided in Appendix C.

---

**Algorithm 1** Our attention mechanism with prompt separation and attention mask adjustment

---

1: **Input:** LLM weight $\theta$, masked prompt $(y_1, \ldots, y_k)$, masked information in encrypted form $(\hat{y}_{k+1}, \ldots, \hat{y}_m)$, number of masks $n$, start positions of masks $(s_1, \ldots, s_n)$, start position of "Masked information:" $t_0$, start positions of each masked information $(t_1, \ldots, t_n)$, lengths of each masked information $(l_1, \ldots, l_n)$, encrypted form of standard causal masks $[\hat{M}_{k+1:m}]$
2: $[Q_{1:k}], [K_{1:k}], [V_{1:k}] \leftarrow p_\theta(y_{1:k})$     ▷ Compute key-value cache for masked prompt (in plaintext)
3: $[\hat{K}_{1:k}], [\hat{V}_{1:k}] \leftarrow [K_{1:k}], [V_{1:k}]$     ▷ Modify keys and values in encrypted form
4: $[\hat{Q}_{k+1:m}], [\hat{K}_{k+1:m}], [\hat{V}_{k+1:m}] \leftarrow \hat{p}_\theta(\hat{y}_{k+1:m})$   ▷ Compute queries, keys, values (in encrypted form)
5: $[\tilde{M}_{k+1:m}] \leftarrow [\mathbf{0}_{k+1:m}]$     ▷ Initialize adjustment mask
6: **for** $i = 1, \ldots, n-1$ **do**
7:   **for** $j = 0, \ldots, l_i - 1$ **do**
8:     **for** $u = s_i, \ldots, t_0 - 1$ **do**
9:       $\tilde{M}_{t_i+j,u} \leftarrow -\infty$
10:     **end for**
11:   **end for**
12: **end for**     ▷ Construct adjustment mask
13: $\overline{[\hat{M}_{k+1:m}]} \leftarrow [\hat{M}_{k+1:m}] + [\tilde{M}_{k+1:m}]$   ▷ Combine causal and adjustment masks
14: $[\hat{H}_{k+1:m}] \leftarrow \text{Attention}([\hat{Q}_{k+1:m}], [\hat{K}_{1:m}], [\hat{V}_{1:m}]; \overline{[\hat{M}_{k+1:m}]})$   ▷ Compute attention (Equation 3)
15: **Output:** $[\hat{H}_{k+1:m}]$     ▷ Output to next layer

---

Specifically, for the tokens of the masked prompt $(y_1, \ldots, y_k)$, we first compute the keys $[K_{1:k}]$ and values $[V_{1:k}]$ in plaintext. Then, when computing the tokens of the masked information $(\hat{y}_{k+1}, \ldots, \hat{y}_m)$ securely, we convert the precomputed keys and values into encrypted form (via secret sharing or homomorphic encryption), denoted as $[\hat{K}_{1:k}]$ and $[\hat{V}_{1:k}]$. We then compute the queries $[\hat{Q}_{k+1:m}]$, keys $[\hat{K}_{k+1:m}]$, and values $[\hat{V}_{k+1:m}]$ for the mask information. Finally, the attention is computed as:

$$\text{Attention}\left([\hat{Q}_{k+1:m}], [\hat{K}_{1:m}], [\hat{V}_{1:m}]\right) = \text{softmax}\left(\frac{[\hat{Q}_{k+1:m}][\hat{K}_{1:m}]^\top + [\hat{M}_{k+1:m}]}{\sqrt{d}}\right)[\hat{V}_{1:m}], \quad (2)$$

where $[\hat{K}_{1:m}] = [\hat{K}_{1:k} \| \hat{K}_{k+1:m}]$ and $[\hat{V}_{1:m}] = [\hat{V}_{1:k} \| \hat{V}_{k+1:m}]$ denote the concatenation of the precomputed and currently computed keys and values, respectively, and $d$ is the dimension of the head. Here, $[\hat{M}_{k+1:m}]$ represents the causal attention mask applied to the mask information tokens. Finally, decoding proceeds as in standard autoregressive inference, using the combined cache.

Although prompt separation might hinder comprehension, recent Large Language Models (LLMs) have demonstrated performance comparable to humans across a wide range of evaluations (Achiam et al., 2023; Team et al., 2024), suggesting that the performance degradation resulting from our prompt separation would be limited. To validate this, we evaluate GPT-4.1 (Achiam et al., 2023) on several subjects within the MMLU benchmark (Hendrycks et al., 2020), such as high-school world history, professional medicine, and professional accounting. These subjects are selected because they frequently contain privacy-sensitive phrases, making them suitable for our evaluation. Note that cases without privacy-sensitive tokens are excluded from the evaluation. Since GPT-4.1 is a closed-source model, we input the combined prompt consisting of the masked prompt followed by the appended masked information. As shown in Table 1, only the history subject shows a performance drop, and it is even less than 0.5%. This result demonstrates that the negative effect of our prompt separation on LLM performance is negligible.

Table 1: Accuracy comparison on MMLU.

| Method (%) | History | Medicine | Account. |
|---|---|---|---|
| GPT-4.1 | 92.80 | 90.46 | 48.00 |
| GPT-4.1+**Ours** | 92.37 | 91.60 | 48.00 |

### 3.2 ATTENTION MASK ADJUSTMENT

To ensure the appropriate attention of privacy-sensitive tokens, we adjust their attention mask. Inspired by the human tendency to reread earlier parts of a text when additional information about a difficult or ambiguous word is provided later in the prompt, we guide the model to refocus on

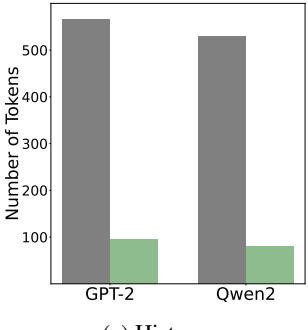 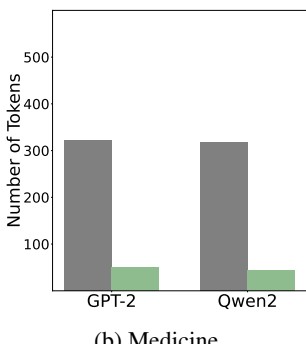 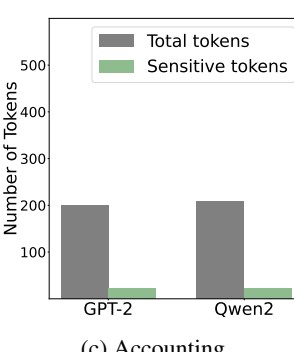

| (a) History | (b) Medicine | (c) Accounting |

Figure 3: Average token counts for the complete prompt and privacy-sensitive tokens on MMLU. Gray bars represent the total number of tokens in the combined prompt (masked prompt + mask information), while green bars represent the number of privacy-sensitive tokens only. 'GPT-2' and 'Qwen2' indicate results obtained using the respective tokenizer and experimental settings.

relevant contextual cues. However, due to the lost-in-the-middle phenomenon (Liu et al., 2023), the model may fail to attend to the original surrounding context of a masked phrase, especially when it appears in the middle of a long sequence. To address this, as illustrated in Figure 2, we constrain the attention masks of privacy-sensitive tokens so that they attend only to the preceding tokens in the original prompt (before their original position) and to the preceding parts of the appended masked information. This is necessary because when attending to tokens before their original position, some of those tokens may have been masked. Thus, allowing access to the preceding portions of the masked information ensures the model can properly reference the missing context. For example, given the masked prompt "The substitution of [MASK_PERSON_0] for the scholastic [MASK_PERSON_1] was hastened" and the appended mask information "Masked information: [MASK_PERSON_0]=Plato, [MASK_PERSON_1]=Aristotle", the tokens corresponding to "[MASK_PERSON_0]=Plato" can only attend to "The substitution of" and "Masked information:". Similarly, the tokens for "[MASK_PERSON_1]=Aristotle" can attend to "The substitution of [MASK_PERSON_0] for the scholastic" and "Masked information: [MASK_PERSON_0]=Plato,".

Specifically, for the $i$-th mask "[MASK_$type\_i$]", let $s_i$ be the original starting position of the $i$-th masked phrase in the masked prompt, and let $t_0$ be the starting position of "Masked information:". Then, for the adjustment mask $\tilde{M}$ for the tokens of the $i$-th mask information, we assign a large negative value for the tokens in the masked prompt from index $s_i$ up to (but not including) $t_0$, and 0 otherwise. Note that the original causal masks have 0 for these tokens. This mask adjustment promotes exploitation of the nearby token information around the original position of the masks. Then, the attention for the masked information is computed as:

$$\text{Attention}\left([\hat{Q}_{k+1:m}], [\hat{K}_{1:m}], [\hat{V}_{1:m}]\right) = \text{softmax}\left(\frac{[\hat{Q}_{k+1:m}][\hat{K}_{1:m}]^\top + \overline{[\hat{M}_{k+1:m}]}}{\sqrt{d}}\right)[\hat{V}_{1:m}], \quad (3)$$

where $\overline{[\hat{M}_{k+1:m}]} = [\hat{M}_{k+1:m}] + [\tilde{M}_{k+1:m}]$ is the summation of a standard causal mask and our adjustment. We note that, for the tokens of the last masked phrase, we use the standard causal mask to encourage the model to consider all preceding information for a comprehensive understanding of the prompt. The detailed algorithm for the attention mechanism with prompt separation and attention mask adjustment is provided in Algorithm 1, and the resulting attention changes induced by this adjustment are analyzed in Appendix F.

## 4 EXPERIMENTS

In this section, we evaluate our prompt separation and attention mask adjustment. First, in Section 4.1, we describe the experimental setup and evaluation protocol, including the specific models, MPC frameworks, and datasets. Next, in Section 4.2, we analyze the number of tokens constituting privacy-sensitive information within the complete prompt. Finally, in Section 4.3, we demonstrate the efficiency of our method across various autoregressive language models on the benchmark dataset. To further validate its effectiveness, we conduct experiments under two different MPC frameworks.

Table 2: Average accuracy and latency comparisons on MMLU using CrypTen. 'Acc' and 'Time' denote accuracy (%) and latency (s), respectively. 'Accel' represents the inference speed acceleration of our method compared to the corresponding baseline.

| Method | | History | | | Medicine | | | Accounting | |
| (CrypTen) | Acc | Time | Accel | Acc | Time | Accel | Acc | Time | Accel |
| --- | --- | --- | --- | --- | --- | --- | --- | --- | --- |
| Plain-GPT-2 | 27.97 | 0.08 | | 19.85 | 0.05 | | 26.00 | 0.05 | |
| GPT-2 | 26.69 | 96.10 | **2.38×** | 19.47 | 59.23 | **2.08×** | 26.67 | 40.18 | **1.64×** |
| GPT-2+**Ours** | 27.12 | 40.35 | | 20.23 | 28.39 | | 26.00 | 24.55 | |
| Plain-Qwen2 | 56.78 | 0.09 | | 27.10 | 0.06 | | 30.00 | 0.06 | |
| Qwen2 | 47.46 | 306.22 | **2.51×** | 25.57 | 210.54 | **2.32×** | 28.67 | 145.78 | **1.75×** |
| Qwen2+**Ours** | 47.03 | 122.22 | | 29.39 | 90.91 | | 28.00 | 83.10 | |

Table 3: Average accuracy and latency comparisons on MMLU using SPU. 'Acc' and 'Time' denote accuracy (%) and latency (s), respectively. 'Accel' represents the inference speed acceleration of our method compared to the corresponding baseline.

| Method | | History | | | Medicine | | | Accounting | |
| (SPU) | Acc | Time | Accel | Acc | Time | Accel | Acc | Time | Accel |
| --- | --- | --- | --- | --- | --- | --- | --- | --- | --- |
| Plain-GPT-2 | 27.27 | 3.02 | | 19.08 | 2.37 | | 26.21 | 1.89 | |
| GPT-2 | 27.27 | 517.68 | **2.40×** | 18.70 | 255.43 | **1.91×** | 27.59 | 168.29 | **1.51×** |
| GPT-2+**Ours** | 25.54 | 215.28 | | 18.70 | 133.87 | | 26.90 | 111.24 | |

## 4.1 EXPERIMENT SETUP

**Implementation details**   We use small autoregressive language models such as GPT-2-small (Radford et al., 2019) and Qwen2-0.5B-instruct (Yang et al., 2024), due to resource constraints. For the frameworks, we adopt CrypTen (Knott et al., 2021) and SPU (Ma et al., 2023), using their default configurations, with several approximation modifications applied to CrypTen. The details of these approximation modifications are provided in Appendix D. For evaluation, we use multiple subjects from the MMLU dataset (Hendrycks et al., 2020), which frequently contains privacy-sensitive phrases such as high school world history, professional medicine, and professional accounting. Note that cases without privacy-sensitive tokens are excluded from the evaluation.

**Evaluation protocol**   We follow the official code of Hendrycks et al. (2020), with a few modifications. GPT-2 has a 1024-token limit, and some instances exceed this when combining the question and choices. To address this, we include only one short example before the main question. Additionally, for GPT-2, we truncate the question to a maximum of 640 tokens and each choice to 32 tokens, ensuring that the total does not exceed 768 tokens (accounting for the example). For Qwen2, which supports a much larger context window, we use the full question and choices without truncation. Performance is reported as average accuracy. Latency measurements are conducted using Intel Xeon Gold 6226R CPU with 256GB RAM and NVIDIA RTX A6000 GPUs (48GB VRAM) for CrypTen experiments, and Intel Xeon CPU E5-2660 v3 with 256GB RAM for SPU experiments. For privacy-sensitive entity categories, we include 'PERSON', 'URL', 'PHONE_NUMBER', 'EMAIL_ADDRESS', 'LOCATION', and 'DATE_TIME' as defined in Presidio (Microsoft, 2024). We use the 'en_core_web_sm' tokenizer from spaCy (Honnibal & Montani, 2017) within Presidio. Further experimental details and protocol specifications are provided in Appendix H.

**Baselines**   'Plain-GPT-2' and 'Plain-Qwen2' denote the performance of GPT-2 and Qwen2 under plaintext inference, respectively. In contrast, 'GPT-2' and 'Qwen2' refer to the results obtained under secure inference for each model.

Table 4: Communication cost comparisons on MMLU using CrypTen. 'Time' and 'Byte' denote communication duration (s) and volume (GB), respectively. 'Drop' represents the communication volume reduction of our method compared to the corresponding baseline.

| Method | History | | | Medicine | | | Accounting | | |
|---|---|---|---|---|---|---|---|---|---|
| (CrypTen) | Time | Byte | Drop | Time | Byte | Drop | Time | Byte | Drop |
| GPT-2 | 59.26 | 91.42 | **3.17×** | 30.77 | 44.78 | **3.30×** | 16.50 | 26.70 | **2.77×** |
| GPT-2+**Ours** | 18.32 | 28.81 | | 10.39 | 13.59 | | 8.37 | 9.63 | |
| Qwen2 | 177.62 | 295.56 | **4.11×** | 110.27 | 161.75 | **4.19×** | 64.23 | 109.53 | **3.59×** |
| Qwen2+**Ours** | 55.95 | 71.86 | | 35.52 | 38.58 | | 31.64 | 30.48 | |

## 4.2 PROMPT SEPARATION ANALYSIS

We analyze the proportion of privacy-sensitive tokens in the MMLU dataset after applying our prompt separation method. Experiments are conducted on high school world history, professional medicine, and professional accounting using the tokenizers and experimental settings of both GPT-2 and Qwen2. As shown in Figure 3, privacy-sensitive tokens account for an average of 13.74% of the total prompt length across these subjects, while the remaining 86.26% are non-sensitive tokens. Surprisingly, in the history subject using the Qwen2 tokenizer, up to 451 tokens can be precomputed in plaintext, and this might lead to significant reductions in inference latency. These results suggest that our prompt separation method can substantially reduce the amount of secure computation required, leading to improved inference speed.

## 4.3 PERFORMANCE AND LATENCY COMPARISON

We evaluate our approach across various subjects in MMLU using two model architectures and two secure inference frameworks. As shown in Table 2, under the CrypTen setting, our method achieves a 1.6× to 2.5× speedup in inference across three subjects with both architectures. The speedup increases with the number of non-sensitive tokens that can be processed in plaintext. Specifically, the largest gains are observed in history, followed by medicine and accounting, reflecting the increasing amount of plaintext computation. Despite these gains, the performance of our method remains comparable to both plaintext and fully secure baselines, with less than a 1% drop in accuracy. We note that the performance gap between Plain-Qwen2 and Qwen2 arises from precision differences between CrypTen and float32. CrypTen employs 16-bit fixed-point arithmetic, whereas RMSNorm (Zhang & Sennrich, 2019) is performed in 32-bit floating-point precision. This mismatch causes many values to be represented as zero in CrypTen, leading to a performance drop. A similar trend is observed under the SPU setting (Table 3), confirming that our method accelerates latency without significant performance degradation in most cases. Specifically, we achieve a 1.5× to 2.4× acceleration in inference speed. Note that inference under SPU is slower than CrypTen in our experiments due to hardware differences (SPU runs on CPUs, whereas CrypTen utilizes GPUs). Additionally, performance discrepancies between SPU and CrypTen arise from the different default approximation methods for non-linear operations in each MPC framework. Further experiments on additional datasets and larger models are provided in Appendix A.

We also analyze communication overhead using CrypTen, measuring both communication duration and data volume. As reported in Table 4, our method achieves a substantial reduction in communication cost between 2.8× and 4.2× compared to the baseline methods. The reduction in communication volume is greater for Qwen2 (4×) than for GPT-2 (3×), as Qwen2 utilizes Grouped Query Attention (Ainslie et al., 2023), which significantly reduces the size of keys and values relative to queries. As a result, sharing the precomputed key-value cache accounts for a smaller proportion of the total communication volume in Qwen2 compared to GPT-2, resulting in a greater overall reduction. These results imply that our method to compute only privacy-sensitive tokens securely is effective for reducing communication costs.

## 5 RELATED WORK

For privacy-preserving inference, Fully Homomorphic Encryption (FHE) (Gentry, 2009; Cheon et al., 2017) and Secure Multi-Party Computation (MPC) (Yao, 1986; Evans et al., 2018; Damgård et al., 2012; Goldreich et al., 2019) are frequently used, as they offer both correctness and privacy. Here, correctness means that the output of private inference should closely match the result obtained from plaintext computation, while privacy implies that it is computationally infeasible to gain any additional information from private inference beyond the intended information, such as the encrypted input or partial shares. FHE ensures privacy through encryption with large security parameters, while MPC achieves privacy by distributing partial secrets along with encryption. Recovering the original value from either the encrypted value or the partial shares is computationally infeasible. However, both FHE and MPC have a challenge: the computationally expensive operations (such as bootstrapping in FHE) and communication costs (in MPC) lead to significantly slower inference speeds compared to plaintext computation. In particular, non-linear operations are a major bottleneck, and polynomial approximations are commonly employed to enable efficient computation. These approximations, however, can introduce numerical errors.

To address these limitations, prior work has explored more efficient cryptographic primitives and approximation techniques. CryptoNet (Gilad-Bachrach et al., 2016) pioneered the use of FHE for convolutional neural networks. Hybrid methods that exploit the advantages of both FHE and MPC have been proposed (Juvekar et al., 2018; Reagen et al., 2021; Hao et al., 2022; Pang et al., 2024; Li et al., 2024; Lu et al., 2025). To further reduce client computation and communication costs, non-interactive approaches such as Nexus (Zhang et al., 2025a) have also been introduced. For non-linear operations, piecewise polynomial approximations are widely used (Li et al., 2023; Dong et al., 2023), and Newton or Goldschmidt iterations are applied for inverse and square-root computations. Since transformers are widely used across diverse domains, privacy-preserving approaches for these models (Hao et al., 2022; Li et al., 2023; Zeng et al., 2023; Zhang et al., 2023; Wu et al., 2024; Zimerman et al., 2024; Dong et al., 2023; Li et al., 2024) are explored. MPCFormer (Li et al., 2023) improves efficiency with polynomial approximations for GELU and exponential function in softmax, and mitigates their performance degradation using knowledge distillation. Iron (Hao et al., 2022), Bolt (Pang et al., 2024), and Nimbus (Li et al., 2024) adopt a hybrid approach, enhancing the efficiency of matrix multiplication in FHE and polynomial approximations for non-linear operations such as GELU and softmax. CipherPrune (Zhang et al., 2025b) has proposed an adaptive input reduction method, pruning tokens layer-by-layer to accelerate inference. Recently, Thomas et al. attempts to accelerate inference by sharding tokens or hidden states at the token level, but this approach exposes parts of privacy-sensitive content without cryptographic protection (without secret sharing or encryption), leaving sensitive tokens partially revealed. In contrast, our method cryptographically protects privacy-sensitive tokens, ensuring that no sensitive content is exposed. Despite these advances, inference latency remains significantly higher than in plaintext, and this issue increases the need for further acceleration. Therefore, in this paper, we propose an efficient autoregressive inference, which can be combined with previous works for primitives or approximations, by prompt separation with key-value caching.

## 6 CONCLUSION

We propose a secure inference method for autoregressive language models based on prompt separation and attention mask adjustment. Specifically, inspired by the observation that not all tokens in a prompt are privacy-sensitive, we identify and mask these tokens. We then precompute the key-value cache for the masked prompt and process the sensitive tokens securely using the cached representations. To ensure appropriate attention for sensitive tokens, we adjust the attention mask to prevent them from attending to tokens beyond their original position. Our experiments show that our method achieves a 1.5–2.5× speedup in inference without a significant performance drop with several architectures under various MPC frameworks.

**Limitations**  Due to the resource constraints, our evaluation is limited to relatively small language models such as GPT-2 (0.1B) (Radford et al., 2019) and Qwen2 (0.5B–1.5B) (Yang et al., 2024). In the future, if the frameworks are improved and more resources are provided, we will evaluate our method with larger language models.

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

## A    ADDITIONAL EXPERIMENTS

To further validate the effectiveness of our method, we conduct an additional experiment on the SAT-English subset of AGIEval (Zhong et al., 2024) using Qwen2-0.5B (Yang et al., 2024) with CrypTen (Knott et al., 2021) and on professional accounting using Qwen2-1.5B. Since SAT-English passages are considerably longer than those in other subjects and the maximum token length of GPT-2 (Radford et al., 2019) is limited to 1024, we evaluate our method only with Qwen2. For the larger Qwen2-1.5B model, we evaluate our method on Accounting, which has relatively shorter sequences, using an NVIDIA A100 SXM GPU (80GB VRAM) to accommodate its higher memory requirements. As shown in Table 5, for the additional dataset, our method consistently accelerates inference without performance degradation, achieving a 5.01× speedup and a 7.01× reduction in communication cost. Similarly, in Table 6, for the larger Qwen2-1.5B model, we observe a 1.71× speedup and a 3.32× reduction in communication cost. These results further demonstrate that our approach enables efficient and scalable secure inference for autoregressive language models. Note that the performance gap between Plain-Qwen2 and Qwen2 arises from precision differences between CrypTen and float32. CrypTen employs 16-bit fixed-point arithmetic, whereas RMSNorm (Zhang & Sennrich, 2019) is performed in 32-bit floating-point precision. This mismatch causes many values to be represented as zero in CrypTen, leading to a performance drop.

Table 5: Average accuracy, latency, and communication cost comparisons on SAT-English with Qwen2-0.5B using CrypTen. 'Comm. Volume' denotes communication cost.

| Method | Accuracy (%) | Latency (s) | Comm. Volume (GB) |
|---|---|---|---|
| Plain-Qwen2 | 49.03 | 0.16 | - |
| Qwen2 | 38.35 | 927.85 | 1141.86 |
| Qwen2+**Ours** | 44.66 | 184.90 | 162.96 |

Table 6: Average accuracy, latency, and communication cost comparisons on professional accounting with Qwen2-1.5B using CrypTen. 'Comm. Volume' denotes communication cost.

| Method | Accuracy (%) | Latency (s) | Comm. Volume (GB) |
|---|---|---|---|
| Plain-Qwen2-1.5B | 39.33 | 0.06 | - |
| Qwen2-1.5B | 31.33 | 172.67 | 213.75 |
| Qwen2-1.5B+**Ours** | 32.67 | 101.21 | 64.31 |

## B    EVALUATION WITH BUMBLEBEE

We additionally evaluate our method with BumbleBee (Lu et al., 2025) using GPT-2 (Radford et al., 2019). Due to high memory requirements, we perform experiments only with the relatively small model (GPT-2). We conduct the experiment on Medicine, which contains a moderate prompt length. In Table 7, our method reduces latency by 2.19× without any performance drop. This result implies that our method can reduce latency across different cryptographic protocols without sacrificing performance.

Table 7: Average accuracy, and latency comparisons on Medicine with GPT-2 using BumbleBee.

| Method | Accuracy (%) | Latency (s) |
|---|---|---|
| Plain-GPT-2 | 19.08 | 2.37 |
| GPT-2 | 22.14 | 627.81 |
| GPT-2+**Ours** | 24.05 | 286.83 |

## C    EXAMPLES FOR PROMPT SEPARATION

In this section, we present the prompts used for in-context examples, questions, and answer choices, both with and without our prompt separation method. In Figure 4, a short in-context example from the anatomy subject is included in all prompts before the main question and answer choices. We choose anatomy because the in-context prompts for other subjects are significantly longer.

---

Question 1. Choose the body cavity containing the pituitary gland.
A) Abdominal
B) Cranial
C) Pleural
D) Spinal
Answer: B

---

Figure 4: The in-context example without prompt separation.

Following the in-context example, we provide the question and answer choices, as shown in Figure 5 (an example from the history subject). Most prompts in history, medicine, and accounting contain privacy-sensitive information, such as personal names, specific dates, or locations. After the choices are presented, the prompt ends with the instruction 'Answer:'. The model is then expected to select one of the choices: ' A', ' B', ' C', and ' D'.

---

Question 2. This question refers to the following information.
"In Northern India the existence of separate States at this period is usually little more than a question of words. A zamindar who paid revenue to the Mogul was clearly in a position of dependence, and if he wished to establish a claim to sovereignty, the first step was to refuse, or omit to pay revenue. Such an omission might, however, arise from various other causes, and it is probable that in Rajputana, Central India, and Chota Nagpur there were numerous chiefs and tribes occupying what constitutional lawyers would regard as an anomalous position, sometimes paying the stipulated revenue, sometimes in open rebellion, and sometimes enjoying practical independence because the Mogul authorities found it inconvenient to undertake active measures of coercion." W.H. Moreland, India at the Death of Akbar, 1920
Moreland's description of revenue collection in the Mughal Empire is best seen as evidence for which of the following generalizations?
A) Only people of certain religions were required to pay revenue to the empire.
B) Geographical differences may have influenced which groups pay taxes.
C) Revenue collection was the only source of funds by which the Mughal Empire operated.
D) The case of Rajputana was a typical one in the Mughal Empire.
Answer:

---

Figure 5: The prompt example without prompt separation.

In our prompt separation approach, most privacy-sensitive tokens are masked, and the corresponding information is provided at the end of the prompt, just before the instruction. As shown in Figure 6, since the target question and answer choices are masked, we also mask the term "pituitary gland" and include it at the end of the prompt. This helps guide the model to understand the structure and format of the question.

Similarly, in the target questions and choices shown in Figure 7, multiple tokens related to names, dates, or locations are masked. For efficiency, we reuse the same mask token for repeated phrases.

Question 1. Choose the body cavity containing the [MASK_PLACE_0].
A) Abdominal
B) Cranial
C) Pleural
D) Spinal
Masked information: [MASK_PLACE_0]=pituitary gland.
Answer: B

Figure 6: The in-context example with prompt separation.

Question 2. This question refers to the following information.
"In [MASK_PLACE_0] the existence of separate [MASK_PLACE_1] at this period is usually little more than a question of words. A zamindar who paid revenue to the [MASK_PLACE_2] was clearly in a position of dependence, and if he wished to establish a claim to sovereignty, the first step was to refuse, or omit to pay revenue. Such an omission might, however, arise from various other causes, and it is probable that in [MASK_PLACE_3], [MASK_PLACE_4], and [MASK_PLACE_5] there were numerous chiefs and tribes occupying what constitutional lawyers would regard as an anomalous position, sometimes paying the stipulated revenue, sometimes in open rebellion, and sometimes enjoying practical independence because the [MASK_PLACE_2] authorities found it inconvenient to undertake active measures of coercion." [MASK_PERSON_0], [MASK_PLACE_6] at the Death of Akbar, [MASK_DATE_0] [MASK_PERSON_1]'s description of revenue collection in [MASK_PLACE_7] is best seen as evidence for which of the following generalizations?
A) Only people of certain religions were required to pay revenue to the empire.
B) Geographical differences may have influenced which groups pay taxes.
C) Revenue collection was the only source of funds by which the Mughal Empire operated.
D) The case of [MASK_PERSON_2] was a typical one in [MASK_PLACE_7].
Masked information: [MASK_PLACE_0]=Northern India, [MASK_PLACE_1]=States, [MASK_PLACE_2]=Mogul, [MASK_PLACE_3]=Rajputana, [MASK_PLACE_4]= Central India, [MASK_PLACE_5]=Chota Nagpur, [MASK_PERSON_0]=W.H. Moreland, [MASK_PLACE_6]=India, [MASK_DATE_0]=1920, [MASK_PERSON_1]=Moreland, [MASK_PLACE_7]=the Mughal Empire, [MASK_PERSON_2]=Rajputana.
Answer:

Figure 7: The prompt example with prompt separation.

## D    APPROXIMATION MODIFICATIONS IN CRYPTEN

In CrypTen (Knott et al., 2021), we follow the default setting except for exponential, SiLU, GELU (Hendrycks & Gimpel, 2016), LayerNorm (Ba, 2016), and RMSNorm (Zhang & Sennrich, 2019). Note that $x$ denotes an input value. We adopt a similar piecewise approximation used in Dong et al. (2023) for non-linear operations. For the exponential, the output is computed as:

$$\exp(x) = \begin{cases} 0, & \text{if } x < -256 \\ \left(1 + \frac{x}{256}\right)^{256}, & \text{if } x \geq -256. \end{cases}$$

For SiLU, the output is calculated as:

$$\text{SiLU}(x) = \begin{cases} 0, & \text{if } x < -6 \\ x \cdot \text{sigmoid}(x), & \text{if } -6 \leq x < -6 \\ x, & \text{if } x \geq 6, \end{cases}$$

where we use the default approximation in sigmoid. Similarly, for GELU, the output is acquired as:

$$\text{GELU}(x) = \begin{cases} 0, & \text{if } x < -4 \\ 0.5x \left(1 + \text{erf}\left(\frac{x}{\sqrt{2}}\right)\right), & \text{if } -4 \leq x < 4 \\ x, & \text{if } x \geq 4. \end{cases}$$

Here, erf is the Gaussian error function. In practice, this function is often approximated as: $\tanh\left(\frac{2}{\sqrt{\pi}}\left(x + \frac{11}{123}x^3\right)\right)$, and the hyperbolic tangent is calculated via the sigmoid function as: $2 * \text{sigmoid}(2x) - 1$.

In LayerNorm in GPT-2 (Radford et al., 2019), the inverse square root might suffer from poor approximation quality in CrypTen when the input values are too small or too large. To address this, we preprocess values that fall outside a stable range (particularly those outside $[2^{-3}, 2^3]$) before normalization. Note that LayerNorm satisfies the following property: $\frac{d \cdot x - \text{mean}(d \cdot x)}{\sqrt{\text{variance}(d \cdot x)}} = \frac{(x - \text{mean}(x))}{\sqrt{\text{variance}(x)}}$, which we exploit during preprocessing.

For RMSNorm in Qwen2 (Yang et al., 2024), we apply similar preprocessing. In Qwen2, precision is increased before variance computation to mitigate underflow issues, often causing the variance to collapse to zero. However, since CrypTen uses fixed precision, we instead scale inputs by $2^6$ when the minimum absolute value across the hidden dimension is below $2^{-4}$. We also preprocess inputs falling outside the stable range, in the same manner as in our GPT-2 preprocessing.

## E  THREAT MODELS

We follow standard secure inference protocols under the semi-honest adversary model, assuming that all parties are computationally bounded and follow the protocol, but may attempt to infer additional information from observed data. In our implementation, we adopt two common MPC settings: a two-party protocol in CrypTen (Knott et al., 2021) and a three-party protocol in SPU (Ma et al., 2023). In our setting, both privacy-sensitive tokens and model parameters remain hidden throughout the inference. The only plaintext inputs are non-sensitive tokens, which are intentionally exposed and thus shared with all parties.

## F  ATTENTION ANALYSIS

To further investigate attention behavior under prompt separation, we conduct an empirical attention analysis using GPT-2 (Radford et al., 2019) on History. We measure how much each sensitive token attends to its original preceding tokens (before masking), averaging over all layers and heads. As shown in Table 8, prompt separation alone leads to weakened attention to relevant context. In contrast, with our adjustment, the attention patterns are restored to closely match those of the original prompt.

Table 8: Average attention mass on original preceding tokens of GPT-2.

| Preceding attention sum | 5 tokens | 10 tokens | 20 tokens |
|---|---|---|---|
| GPT-2 | 0.0456 | 0.0913 | 0.1826 |
| GPT-2+**Ours** (w/o adjustment) | 0.0113 | 0.0226 | 0.0453 |
| GPT-2+**Ours** | 0.0403 | 0.0806 | 0.1615 |

## G  PERFORMANCE IN OTHER TYPES OF SENSITIVE INFORMATION

To demonstrate that our method is not limited to a fixed PII detector and can generalize to other types of information, we conduct an additional evaluation where both a sensitive phrase (detected by a PII detector) and a random phrase are masked simultaneously. Specifically, we use spaCy to extract noun phrases (e.g., "the Accounting Standards Codification", "an investment project") that do not overlap with the PII spans. From these, we randomly select one phrase and mask it using '[MASK_0]', without specifying its entity type.

In Table 9, while we observe a slight performance drop in Medicine, the overall degradation remains limited. We attribute this to cases where the randomly selected phrase is task-critical information, making it more difficult for the model to understand the question. Nevertheless, this result indicates

that our method remains effective even when masking arbitrary (non-PII) content, further supporting its generalizability beyond fixed sensitive-token detectors.

Table 9: Average accuracy (%) on History, Medicine, and Accounting with Qwen2 using CrypTen. 'Qwen2 + Ours (w/ additional mask)' denotes performance when an additional random noun phrase is masked alongside the PII span.

| Method | History | Medicine | Accounting |
|---|---|---|---|
| Plain-Qwen2 | 56.78 | 27.10 | 30.00 |
| Qwen2 | 47.46 | 25.57 | 28.67 |
| Qwen2+**Ours** | 47.03 | 29.39 | 28.00 |
| Qwen2+**Ours** (w/ additional mask) | 48.73 | 27.86 | 29.33 |

## H  EXPERIMENTAL SETUP

We now provide a more detailed description of the dataset and the evaluation protocol.

**Dataset details**   The MMLU benchmark dataset (Hendrycks et al., 2020) is designed to evaluate the understanding capabilities of language models across 57 diverse tasks, ranging from elementary-level subjects to professional domains such as medicine, accounting, and law. For our experiments, we focus on subjects that involve privacy-sensitive information. We exclude tasks that primarily assess general knowledge, such as mathematics and science. Instead, we select subjects that involve real-life scenarios or historical contexts, which frequently include names, dates, and locations. Specifically, we use the following subjects for evaluation: high school world history, professional medicine, professional accounting, and professional law.

**Evaluation protocol**   We follow the official code from Hendrycks et al. (2020). In the MMLU dataset, each prompt consists of several example problems, followed by a question and multiple choices. The prompt ends with the instruction 'Answer:', after which the model is expected to predict the correct choice. The prediction is made by computing the log probabilities assigned by the language model to the possible answer options (' A', ' B', ' C', ' D'), and selecting the one with the highest probability. To encourage the model to produce answers in the choice format, it is common to include five in-context examples related to the subject. However, due to model and resource constraints, we use only a single example. This example is deliberately kept short and is not related to the subject of the target question, since the subjects we selected often have long questions and choices.