# OpenReview forum: "Secure Autoregressive Inference with Prompt Separation via Key-Value Caching"
_ICLR.cc/2026/Conference — Submitted to ICLR 2026_

### Official Review · Reviewer_F5HB · 2025-10-21

**Soundness:** 2
**Presentation:** 2
**Contribution:** 2
**Rating:** 4
**Confidence:** 4

**Summary:**

Traditional MPC reasoning requires re-execution of security protocols after each token is generated, resulting in extreme latency. This work addresses the token-by-token dependency issues of auto-regressive LLMs by implementing structured predictions. This involves performing computations on non-sensitive tokens on the plaintext side first, calculating partial activations without leaking previous tokens. Furthermore, the attention buffer is secretly shared, eliminating the need for re-negotiation of keys during cross-stride reasoning, reducing communication traffic and encryption costs.

**Strengths:**

The research problem is valuable, and the solution is highly compatible with existing MPC frameworks (such as CrypTen and SecFormer). The method is tested on **GPT-2** and **Qwen2-0.5B-Instruct**, and under two MPC frameworks, **CrypTen** and **SPU**, respectively, achieving **1.5–2.5×** inference acceleration on several MMLU subtasks with very low accuracy loss.

**Weaknesses:**

**1.Idealistic security assumptions:**

The security proof is incomplete. The authors assume that "non-sensitive tokens are visible in plaintext," but overlook the fact that the length, structure, or position pattern of plaintext prompts can also lead to side-channel information leakage.

**2.Limited parallelism:**

Although the scheme reduces communication rounds, inference parallelism is still constrained by the depth of autoregressive dependencies, making it difficult to completely eliminate the sequencing bottleneck.

**3.Lack of systematic comparative experiments:**

While the paper mentions related work (MPCFormer, Bolt/Iron, etc.), it lacks cross-sectional experiments using the same dataset and framework, or overlay tests with other protocols.

**Questions:**

1. Are the structure, length, and position features of the plaintext considered potentially private information?

2. Can the KV be reused across requests after being converted from the plaintext domain to the ciphertext domain?

3. Does it maintain near-zero performance with long contexts (>32k), enhanced retrieval (RAG), conversational security (rejection policy), and code generation?

---

> ### Author Response · Authors · 2025-11-20
>
> We deeply appreciate the reviewer’s constructive input and the care taken in evaluating our work.
>
> ---
>
> ### **Potential threat**
>
> We agree that, since we do not treat all tokens as private by default, certain contextual cues may remain visible and could, in principle, aid inference about hidden content. We therefore do not claim the same level of end‑to‑end privacy as works that encrypt the entire input. However, to prevent revealing these cues, we allow **users can specify additional spans as sensitive, such as domain terms, or noun phrases,  so that those tokens are also processed securely**. In Appendix G, we mask random noun phrases in addition to PII and observe comparable performance with only slight degradation, demonstrating that our method continues to work when the sensitive set is broadened beyond PII. Therefore, our approach offers a meaningful option for accelerating secure inference in scenarios where contextual cues are not critical, providing a flexible balance between efficiency and privacy.
>
> ---
>
> ### **Inference parallelism**
>
> We would be happy to address the reviewer’s concern once the notion of inference parallelism is specified more concretely.
> **If the reviewer refers to batch parallelism, our method introduces no additional bottleneck compared to existing approaches and is, in fact, more favorable in this setting**. In typical batch inference, multiple samples are processed simultaneously. Since their sequence lengths differ, all samples must be padded to match the longest one, which increases computation and communication overhead, particularly when one sample is significantly longer than the others.
>
> In our approach, however, the input is separated into non-sensitive and sensitive segments, and only the sensitive portion requires secure computation. Since the sensitive spans constitute roughly 1/8 of the total tokens, the variance in sequence length across samples becomes substantially smaller when considering only the sensitive part. As a result, padding-based inefficiencies and thus unnecessary communication costs are expected to be significantly reduced.
>
> ---
>
> ### **Combination with other protocols**
>
> We also conducted experiments using BumbleBee [1], a recent hybrid approach (similar to Iron and Bolt). As reported in Appendix B, we evaluated our method on Medicine with GPT-2 and observed results consistent with our main results.
>
> As the reviewer suggested, we already attempted to extend these experiments to other datasets and models, such as History and Qwen2. However, this was not feasible under our available computational resources. In this dataset, input prompts can exceed 1000 tokens, and the encrypted vectors become significantly larger under homomorphic encryption according to security parameters.
> Consequently, the memory usage grows substantially, and with our current 256 GB memory limit (even with 480 GB RAM), we were unable to run BumbleBee for longer prompts and larger models.
>
> [1] Lu, Wen-jie, et al. "Bumblebee: Secure two-party inference framework for large transformers." NDSS, 2025.
>
> ---
>
> ### **Potential leakage from meta information**
>
> First, we note that the token length of an individual sensitive phrase is not observable since we replace every sensitive span with a mask of a fixed form, regardless of its original length.
> We also believe that the amount of information that can be inferred solely from structural or positional cues is limited.
>
> ---
>
> ### **Reuse of KV cache after converting to the ciphertext domain**
>
> Depending on the protocol, it may not be strictly necessary to convert the KV cache into the ciphertext domain. However, under the MPC setting used in our experiments, the KV cache itself could potentially reveal information about the model to the client. For this reason, we performed the experiments by secret-sharing the KV cache, ensuring that no model-dependent information is leaked while still enabling correct reuse.
>
> ---
>
> ### **Performance when extending to the applications**
>
> At present, secure inference remains highly demanding in terms of both latency and memory consumption, making it difficult to meaningfully evaluate our method under the scenario suggested by the reviewer. Once such experiments become feasible in practice, we view this as a valuable direction and plan to explore it as future work.

---

### Official Review · Reviewer_x5aM · 2025-10-31

**Soundness:** 2
**Presentation:** 3
**Contribution:** 3
**Rating:** 6
**Confidence:** 4

**Summary:**

This paper proposes a prompt separation technique to accelerate secure Transformer inference based on Fully Homomorphic Encryption (FHE) and Secure Multi-Party Computation (MPC). The core insight is that not all input tokens carry privacy-sensitive information. Leveraging this, the authors separate private and non-private tokens using a Personally Identifiable Information (PII) identifier. The key-value (KV) cache for non-private tokens, along with their corresponding masks, is computed in plaintext and reused during secure inference over private tokens. This avoids computing the entire KV-cache in the encrypted domain. Experimental results demonstrate that the proposed approach can accelerate inference by up to 2.5 times.

**Strengths:**

1. The paper is well written and easy to follow. The prompt separation framework and its integration with KV caching are clearly explained.

2. The research topic is important and well-motivated. Enhancing the efficiency of secure Transformer inference has direct relevance to privacy-preserving real-world applications.

3. The evaluation is thorough. The authors conduct experiments on both GPT-2-small and Qwen2-0.5B-Instruct, showing consistent speedups over baseline methods.

**Weaknesses:**

1. The threat model requires further clarification:

- This work assumes that only certain tokens—those identified as PII (e.g., PERSON, PHONE_NUMBER, DATE_TIME)—are privacy-sensitive. This contrasts with prior works such as MPCFormer, Iron, and Bolt, which treat the entire input as private. However, contextual information like grammar, tense, or sentence structure may also leak private information. Clarifying the scope of what is considered private would make the threat model more rigorous.

- Additionally, the claim in line 831 that model parameters remain hidden even in the two-party setting is inaccurate for frameworks such as Iron and Bolt, where model weights are held in plaintext by the server.

2. There is noticeable accuracy degradation. Although the proposed technique improves inference efficiency, it sometimes leads to reduced accuracy, as shown in Tables 2 and 3. For instance, on the History dataset using the SPU backend, accuracy drops from 27.27 to 25.54.

**Questions:**

1. Are there known failure cases of the prompt separation approach? Would fine-tuning the LLM in a mask-aware manner help mitigate the accuracy loss?

2. How is privacy leakage assessed when non-private tokens are exposed to an untrusted server? Are there quantitative metrics or threat analyses that support the claimed privacy guarantees?

3. The current evaluation is limited to relatively small models such as GPT-2-small. How does the proposed technique scale to larger models in terms of performance and security trade-offs?

---

> ### Author Response · Authors · 2025-11-20
>
> We sincerely appreciate the reviewer’s thoughtful feedback and are grateful for the opportunity to deepen the discussion and improve our work.
>
> ---
>
> ### **Privacy concerns**
> We agree that, since we do not treat all tokens as private by default, certain contextual cues (e.g., grammar, tense, local syntax) may remain visible and could, in principle, aid inference about hidden content. We therefore do not claim the same level of end‑to‑end privacy as works that encrypt the entire input (e.g., MPCFormer, Iron, Bolt). To make the threat model precise, we will clarify the default scope and extensibility. By default, the private set only includes PII categories. These tokens are computed securely, but non-PII tokens are shared.
> However, our method extends beyond PII. We allow **users can specify additional spans as sensitive, such as domain terms, or noun phrases, so that those tokens are also processed securely**. In Appendix G, we mask random noun phrases in addition to PII and observe comparable performance with only slight degradation, demonstrating that our method continues to work when the sensitive set is broadened beyond PII. We clarify this scope in our revision.
>
> ---
>
> ### **Threat model clarification**
>
> We thank you for pointing out the imprecise wording. Our experiments were conducted in vanilla MPC configurations, where both the model parameters and inputs are secret‑shared. Our original statement was intended to describe this MPC setting, not to generalize across all secure‑inference frameworks. However, our method itself is framework‑agnostic, so when combined with hybrid approaches (Iron or Bolt), we do not claim parameter secrecy. We will update the text to reflect this scope precisely.
>
> ---
>
> ### **Accuracy drop**
>
> We acknowledge the observed decrease on History. However, we observe that lower‑capacity models such as GPT‑2 are more sensitive to the reordering step and can exhibit modest degradation. In contrast, with stronger models (Qwen2), accuracy is comparable in nearly all cases, and although not measured under MPC, GPT‑4.1 shows negligible degradation under the same reordering protocol. Since real‑world deployments typically use models at least as capable as Qwen2, we expect the practical accuracy impact to be less significant.
>
> ---
>
> ### **Failure cases and finetuning**
>
> We thank you for the helpful suggestion. We are currently analyzing failure cases and will upload the results. For finetuning, mask-aware adaptation would be helpful for mitigating the accuracy drop. However, with stronger backbones (Qwen2 and GPT‑4.1), we already observe comparable or negligible loss with our method, making fine‑tuning optional rather than required.
>
> ---
>
> ### **Assessing leakage with exposed non‑private tokens**
>
> We are designing and running an evaluation of context leakage and will report the results as soon as they are available.
>
> ---
>
> ### **Scaling models and privacy trade-off**
>
> In Appendix A, we scale up our experiments to Qwen2-1.5B for Accounting and observe the same trend reported in the main paper. We were unable to evaluate larger models primarily due to memory constraints. Specifically, the dataset contains sufficiently long inputs with privacy-sensitive tokens, and depending on the sample, the input length can exceed 1,000 tokens. This makes secure inference with larger models infeasible under our available resources.
>
> Regarding the privacy trade-off, it is less dependent on model scaling and more determined by how many tokens are masked. As noted in a previous question, if we quantify the threat model precisely, we will be able to provide a more concrete privacy analysis. We will share the corresponding results once the evaluation is completed.

---

> > ### Author Response · Authors · 2025-12-03
> >
> > We provide further experimental results and analyses below.
> >
> > ---
> >
> > ### **Failure cases**
> > The PII detector we currently use is rule-based and therefore not fully accurate. However, using a small LM–based detector could yield significantly more reliable PII identification. Employing such a small LM on the client side would incur only minimal overhead compared to running full LLM inference.
> >
> > For failure cases, our current rule-based detector also tends to over-detect PII. In particular, it often misclassifies tokens containing a period (‘.’) as names or locations, leading to unnecessary masking of non-sensitive content.
> >
> > ---
> >
> > ### **Assessing leakage with exposed non‑private tokens**
> > To quantify the privacy-leakage risk of our masking strategy, we measure the negative log-likelihood (NLL) of predicting a private token using only non-sensitive tokens. Assuming the true token length is known (which our method does not reveal), we replace the private span with the corresponding number of BERT mask tokens and use BERT-Large to estimate the NLL of predicting the first masked token. We use BERT instead of an autoregressive model to fully leverage bidirectional context. Because NLL varies with token length, we conservatively treat predicting the first token as a successful reconstruction. As shown in the table, BERT’s vocabulary size is 30,522, so random guessing yields 10.3262. For Medicine, our strategy results in 9.3666, approximately three times higher probability than random guessing, yet still extremely small in absolute terms. Furthermore, in practice, the true token length is unknown, and even correctly guessing the first token provides no guarantee of predicting subsequent tokens, making actual reconstruction probability even lower. Overall, these results indicate that **our masking strategy preserves privacy at a level close to random guessing**.
> >
> > | NLL | History | Medicine | Accounting |
> > |---|---|---|---|
> > | Random guess | 10.3262 | 10.3262 | 10.3262 |
> > | Ours  | 10.5596 | 9.3666 | 9.9931 |

---

### Official Review · Reviewer_HCNH · 2025-11-01

**Soundness:** 2
**Presentation:** 2
**Contribution:** 2
**Rating:** 2
**Confidence:** 5

**Summary:**

This paper presents an interesting idea to accelerate the performance of secure inference over private tokens by separating the private tokens from the rest of the prompt. By only hiding the computation over the private tokens, they reduce the amount of operations that needs to be performed within the secure computation.

**Strengths:**

The idea presented in this work is novel and has the potential to substantially improve the performance of semi-private inference.

**Weaknesses:**

While the idea in this paper is very interesting, I was not able to verify correctness due to a lack of explanation for the other parts of the attention block. The matrix products in the attention block mix the values across tokens, so it is not clear from the paper how these values are handled. In particular, if the intermediate states that are functions of the private tokens are not revealed, then this approach seems like if would be nullified after the first block (since the entire state would be a function of the private tokens). On the other hand, if the intermediate results are revealed except at the locations of the private tokens, this could leak information about the private tokens. More explanation is needed on these steps.

**Questions:**

I would like to understand how the matrix operations within the attention blocks are performed with some of the input rows masked. It seems like most (if not all) of the entries of the final output of the attention block are functions of all input tokens, so if some of these values are masked then it’s not clear how this output is computed for the subsequent blocks. Could you give a complete description of the modified attention block (including all matrix operations) with the masked tokens?
When masking private tokens, is there a generic embedding for each category? Or is the embedding for the private tokens replaced with whatever embedding the model assigns to the token “[MASKED_TOKEN_#]”?

---

> ### Author Response · Authors · 2025-11-20
>
> We truly appreciate the reviewer’s thoughtful and constructive feedback.
>
> ---
>
> ### **Detailed description for our attention mechanism**
>
> Our method targets decoder-only autoregressive language models such as GPT and LLaMA. Unlike BERT-style encoder–decoder architectures, there is no separate encoder that can attend to all tokens irrespective of position. Instead, even when processing the input prompt, the model uses a causal mask (as illustrated on the left of Figure 2). Each position $i$ can attend only to positions $\le i$, i.e., only to previous (and itself), not to future tokens. In terms of the attention mask $M$, for each position $i$, all entries corresponding to positions $j>i$ are set to a large negative value so that the softmax attention to future tokens becomes zero. This causal structure is applied in every attention block.
>
> Specifically, the attention block first computes $Q,K,V$ from the hidden states, and then forms attention weights as $\frac{QK^\top}{d}$, where $d$ is the head dimension. In standard decoder-only models, the causal mask is incorporated as $A=\frac{QK^\top + M}{\sqrt{d}}$. As described above, entries corresponding to $j>i$ receive large negative values while allowed positions receive 0, as shown on the left of Figure 2 (green = 0, gray = large negative). In our method, we additionally apply the mask shown on the right of Figure 2, which also assigns large negative values between the original positions of sensitive tokens and the locations where their placeholder declarations appear. After masking, the model computes $AV$. This process repeats in every subsequent attention block.
>
> For behavior across blocks and privacy implications, due to the causal mask, tokens at earlier positions never attend to later positions, in any layer. In the first block, the hidden state at an early position depends only on the prefix (its own position and earlier tokens) and, in the plaintext view, the placeholder embeddings at sensitive positions (not on the true private token values). After the first block, these intermediate vectors propagate to the next block, but the same causal mask is applied again. Therefore, even in deeper layers, the hidden state at an early (non-sensitive) position never incorporates information from later (sensitive) tokens. As a result, intermediate representations at non-sensitive prefix positions remain functions only of non-sensitive tokens and the fixed placeholder embeddings, even after multiple layers. They do not become functions of the true private tokens. Computations that involve the true private tokens (their actual embeddings and any states causally influenced by them) are carried out in the secure domain.
>
> For masking sensitive tokens (input replacement), we do not introduce a separate generic embedding for each category. Instead, we follow the second option mentioned in your question: We literally replace each sensitive token in the plaintext view with a placeholder token such as "[MASK_PLACE_0]", "[MASK_PLACE_1]", etc. These placeholders are then processed by the original tokenizer and embedding layer of the base model. Thus, the embedding at a masked position is exactly the embedding that the pretrained model assigns to the tokenized form of "[MASK_PLACE_0]". Thus, we do not introduce any new embedding parameters.

---

### Official Review · Reviewer_d7wb · 2025-11-02

**Soundness:** 3
**Presentation:** 3
**Contribution:** 3
**Rating:** 6
**Confidence:** 3

**Summary:**

This work proposes a method to accelerate privacy-preserving inference for LLM under homomorphic encryption through a technique that separates sensitive and non-sensitive tokens in a prompt. Then, non-sensitive tokens are processed in plaintext to precompute key-value caches, while sensitive tokens are processed securely using these cached representations. An attention-mask adjustment mechanism ensures that sensitive tokens still attend to relevant context. Experiments on GPT-2 and Qwen2 under MPC frameworks show 1.5–2.5× faster inference with minimal performance loss, and communication costs drop up to 4×. This approach maintains security while making encrypted inference substantially more practical

**Strengths:**

The proposed framework of processing the majority of the insensitive text in plain text and only processing the sensitive information in cyphertext is quite ingenious. While there is still a security concern in terms of whether this protocol can really protect one's privacy (redacted documents often do reveal a lot of private information), given the need for some degree of security while not incurring an inordinate compute cost is a nice compromise.

**Weaknesses:**

The attention mask adjustment feels unnecessary to me, and it doesn't seem like the paper provides sufficient evidence supporting the necessity of this mechanism.

Also, I was hoping the acceleration would be larger than 2x or 4x. Can the authors explain why the speedup is not more extreme despite the sensitive words only consisting of a small portion of the input prompt?

**Questions:**

What evidence do you have that the proposed attention sink mechanism is necessary? Is this really the case?

---

> ### Author Response · Authors · 2025-11-20
>
> We sincerely thank the reviewer for the thoughtful remarks and constructive guidance.
>
> ---
>
> ### **Validation about mask adjustment**
>
> We agree that sufficiently strong models may perform well even without mask adjustment. However, weaker models struggle to leverage the original contextual cues after reordering, which motivates our mechanism. We provide evidence in Appendix F. Specifically, on GPT‑2, we measure, for each layer, how much the token at the reveal position of a sensitive token attends to its original preceding context. As reported in Table 8, without mask adjustment, the average attention to the original preceding context drops by roughly one‑third. This indicates that the model fails to properly condition on the intended local context after reordering. With our mask adjustment, the attention to the original context recovers to near the original‑prompt level, demonstrating that the mechanism effectively preserves the intended positional and structural cues.
>
> ---
>
> ### **Insufficient acceleration**
>
> To evaluate performance faster, we did not throttle network speed in our experiments. This choice is conservative and unfavorable to our method. In realistic deployments with constrained links, such as a LAN (3 Gbps, 0.8 ms RTT) or WAN (100 Gbps, 80 ms RTT), the communication cost becomes the bottleneck, and end‑to‑end latency scales approximately with communication. Since Table 4 shows a 3-4× reduction in communication, we expect a comparable 3-4× latency reduction in such a setting.
>
> As for why the speedup does not scale directly with the fraction of sensitive tokens (1/7), our approach does not remove the non‑sensitive prefix from attention. We compute the non‑sensitive prefix in plaintext and then secret‑share its KV cache so that the secure computation over sensitive tokens can still attend over the entire prefix. Consequently, there remains a cost to secret‑share the prefix KV caches, and for each sensitive token, the secure attention still performs $QK^\top$ over the values of non‑sensitive tokens and consumes communication proportional to that length. These two factors create a speedup floor that prevents linear scaling with the sensitive‑token fraction. Empirically, this yields ~4× communication reduction rather than the naive ~7× implied by a 1/7 sensitive share.

---

### Comment · Area_Chair_jUdd · 2025-11-28
**A gentle reminder to participate in the author–reviewer discussion.**

Dear Reviewers,

Thank you once again for your service to ICLR 2026. Now that the authors have submitted their rebuttal, could you please engage in the interactive discussion with them? Your participation would be very helpful to the authors, and they would greatly appreciate it. Please also read the authors’ response together with the other reviews and consider whether the rebuttal or any additional comments influence your assessment of the paper.

Thank you again for your efforts.

Best wishes,

Your AC

---

### Meta-Review · Area_Chair_Qnh3 · 2025-12-26

**Summary:**

This paper proposes a new technique make large language models (LLMs)  run faster when they keep some parts of the input private. The main idea is to split the prompt into two types of words:
1. Non-private ones (like common words or structure) — these can be processed normally and quickly.
2. Private ones (like personal info or sensitive details) — these are handled with extra security (using techniques like secure multi-party computation, or MPC, which hides the data from the server).

The technique is to process the non-private words first, save their results in KV cache, and then reuse those saved results when securely processing the private words. They also modified how the model pays attention to different words so the order does not break the logic.
They tested it on smaller models (GPT-2 and Qwen2-0.5B) and observed speedups of about 1.5–2.5 times with only a small drop in accuracy.

The paper has some critical issues. **Reviewers found the privacy protection unclear, the security reasoning hard to follow, the speed gains not impressive enough, and the tests limited.**

**Reviewer Concerns:**

Concerns Shared Reviewers:

1. The Privacy Protection is NOT Strong. The proposed technique only hides certain specific words (e.g., names, addresses), but everything else — including sentence structure and surrounding words — is shown openly. Reviewers said this could still leak private data (for example, the way sentences are built might hint at personal details). The authors said users can mark more words as private, but this makes the original idea feel weak compared to methods that treat the whole prompt as private.

2. It is Hard to Tell If the Method Is Actually Secure. Reviewers cannot confirm that the model works properly when mixing public and private parts. They worried that hidden private words will still affect the results in secret ways (leaking privacy).  **The authors cannot answer this question or provide convincing data**.

3. The Speedup Is NOT significant. The speedup is only 1.5–2.5 times, even though private words are usually a small part of the prompt. Reviewers felt this is NOT worth the added complexity (special masking, cache management, etc.). The authors said bigger gains would show up in real-world setups with slow networks, but **they did NOT actually test that**

4. Tests Are Limited. The authors only used small models and simple tasks. Reviewers wanted to see how it works on bigger models, longer conversations, or more realistic uses. The authors said they would do that later, but **now the paper feels incomplete and preliminary**.

Reviewer-Specific Questions:

1. One reviewer (HCNH) seriously doubted if the method was even correct, because attention can mix information across all words in ways that might ruin the separation.

2. Others (d7wb, F5HB) questioned why the special attention tweak was needed and noted the speedup was smaller than hoped. They also worried about side-channel leaks (like the length of the prompt giving hints).

3. Another (x5aM) said this approach is very different from previous secure AI methods and that the accuracy drop is noticeable in some cases.

**Reviewer Scores:**

The reviewer scores are reasonable.

---

### Decision · Program_Chairs · 2026-01-26

Reject